# Recent Advances in Single-Molecule Sensors Based on STM Break Junction Measurements

**DOI:** 10.3390/bios12080565

**Published:** 2022-07-26

**Authors:** Shan-Ling Lv, Cong Zeng, Zhou Yu, Ju-Fang Zheng, Ya-Hao Wang, Yong Shao, Xiao-Shun Zhou

**Affiliations:** Key Laboratory of the Ministry of Education for Advanced Catalysis Materials, Institute of Physical Chemistry, Zhejiang Normal University, Jinhua 321004, China; lslzjnu@163.com (S.-L.L.); zengc20000801@163.com (C.Z.); yuzhou1213@163.com (Z.Y.); jfzheng@zjnu.cn (J.-F.Z.); yshao@zjnu.cn (Y.S.)

**Keywords:** single-molecule sensor, STM break junction, ionic detection, pH detection, nucleotide detection

## Abstract

Single-molecule recognition and detection with the highest resolution measurement has been one of the ultimate goals in science and engineering. Break junction techniques, originally developed to measure single-molecule conductance, recently have also been proven to have the capacity for the label-free exploration of single-molecule physics and chemistry, which paves a new way for single-molecule detection with high temporal resolution. In this review, we outline the primary advances and potential of the STM break junction technique for qualitative identification and quantitative detection at a single-molecule level. The principles of operation of these single-molecule electrical sensing mainly in three regimes, ion, environmental pH and genetic material detection, are summarized. It clearly proves that the single-molecule electrical measurements with break junction techniques show a promising perspective for designing a simple, label-free and nondestructive electrical sensor with ultrahigh sensitivity and excellent selectivity.

## 1. Introduction

Single-molecule sensors with extreme detection limits and the ability to reveal heterogeneity and stochastic processes [1,2], have attracted widespread attention [1] in chemical, physical, and biological sciences. During the past decades, some emerging methods have been developed to allow single-molecule measurements with sufficient speed and statistical accuracy, such as nanopore [3,4,5], microfluid [6,7], and single-molecule fluorescence microscopy [2,8]. It has been found that molecular counting might be the most accurate method for single molecule detection and analysis [9].

Recently, scanning tunneling microscopy-break junction (STM-BJ) [10,11], invented to measure electron transport by repeatably forming single-molecule junctions in a nanogap between two electrodes, has also been a unique platform for exploring the intrinsic properties of materials and the interaction of individual molecules at a single-molecule level [12,13,14]. The tunneling currents in the molecular junctions are sensitive to molecular structure and configuration, interfacial coupling between the anchoring group and electrode, external stimulus and the surroundings [15,16]. Therefore, the transduction features of tunneling current, single-molecule conductance peak and values can also be applied in designing a single-molecule sensor [15]. Compared with traditional single-molecule detections based on optical, physical and chemical methods, such as fluorescence [17,18], surface-enhanced Raman spectroscopy [19,20,21,22], and nanopores [23,24], the electrical sensors based on single-molecule conductance measurements can be complementary and has unique advantages with integrating the STM technique: (1) It is a label-free real-time electrical detection that requires only a small amount of sample. (2) It is not limited by the working environment. STM can efficiently work in vacuum, atmosphere, solution and other different environments. (3) Dynamic and static molecular information at the interfaces, such as molecular adsorption and surface reaction, can be detected in a real space.

From this perspective, we review the primary advances of single-molecule sensors based on break junction techniques in three regimes: ion detection, environmental pH, genetic material detection. The typical examples of molecular junctions constructed with STM-BJ illustrated in this review for the three types of sensors are summarized in Table 1. The principles of operation for this single-molecule electrical recognition, as well as the primary strategies for quantitative detection, are discussed in the following three sections. Compared to ensemble measurements in many traditional analytical methods, the single-molecule measurement by the break junction technique can not only provide detections with the ultimate resolution, but also is capable of discovering the wealth of molecular information hidden in conventional ensemble experiments. On the other hands, there is still a long way for the applications of the single-molecule conductance measurements in real detecting. The current studies are a proof-of-concept in the laboratory. A brief conclusion and outlook appear in the end.

## 2. STM Break Junction Technique

STM-BJ has become a powerful tool to study the unique features of electron transport at a single-molecule level since being invented in 2003 by Tao and co-authors [10]. It is one of the most commonly used techniques and plays a key role in the field of molecular electronics [11,38,39,40]. Typically, this in situ break junction technique constructs thousands of metal–molecule–metal junctions by using a metallic STM tip as one electrode and a metallic substrate covered with target molecules as the other electrode. As illustrated in Figure 1, the tip is pushed into contact with the substrate, then retracted at a constant speed (ca. 20 nm/s) via piezoelectric control. During the tip stretching process, metal atomic contact can be formed, then pulled off. Meanwhile, the target molecules with two terminal anchoring groups can bind to the metal atoms in the nanogap to form single-molecule junctions, and the tip tunneling current is recorded for statistically analyzing single-molecule conductance. Typically, the conductance–displacement curves show a short step feature in Figure 1b at *G*_0_ (*G*_0_ = 2*e*^2^/*h*, where *e* is the electron charge and *h* is Plank’s constant) corresponding to the quantum conductance of atomic metal contact (Such as Au), then a long step feature that appears at a later stage is assigned to the formation of single-molecule junctions. Thousands of these curves are used to construct the one-dimensional conductance histograms with conductance peaks, illustrated in the right panel of Figure 1b.

During the past two decades, numerous studies using STM-BJ have systematically investigated the effects of molecular structures [41,42,43,44], electrode materials [45,46,47], and external stimuli [48,49,50] on electron transport through single molecules. Beyond single-molecule conductance measurements, STM-BJ has been recently extended to study single-molecule physics and chemistry including interfacial molecular adsorption and reaction process [12,51,52,53,54] because molecular adsorption on the surface is the essential step in the formation of a molecular junction, and the corresponding conductance–distance traces are sensitive to small changes in atomic and molecular configurations. Then the statistical analysis of these current traces in the characteristic conductance peak intensities, and the values enable a chemical resolution for single-molecule detection with high temporal resolution.

## 3. pH Detection

With single-molecule break junction techniques, numerous investigations have systematically studied the molecular structure, anchoring groups and electrode materials of metal–molecule–metal junctions over the past two decades [55,56,57,58,59]. Gradually, researchers have turned to tune electron transport in single-molecular junctions under an external stimulus for functional electronic components, such as switch [60,61], rectifier [40] and transistor [39,62]. The pH, as one of the most widely used chemical stimuli, has also been studied in the single-molecular junctions [63,64]. It has been found that the conductance peaks and values of pH-sensitive molecular junctions display very different features in acidic and basic solutions, which paves a new way for designing single-molecule pH sensing.

To be capable of responding to the environment pH stimuli, commonly, it requires the constructions of molecular junctions with pH-sensitive molecular structures. A typical example is illustrated in Figure 2a. Li et al. used two dye molecules of malachite green (MG) and pararosaniline (PA) as molecular pH sensing units [25]. The molecules can undergo reversible structural transformation between a conjugated form in slightly acidic solution and a nonconjugated form in a basic solution. The theoretical calculation reveals that changing the hybridization of the central C atom from sp^2^ at pH = 5.5 to sp^3^ at pH = 13.6 can significantly enlarge the HOMO-LUMO gap, which results in the single-molecule conductance in the basic solution being about 100 times less than that in the acid solution (Figure 2b). This proves the concept of single-molecule pH electrical sensor based on the pH-induced conjugation and electronic change of dye molecules.

Peptides contained pH-sensitive amino and carboxyl groups have also been explored at the single-molecule level by break junction techniques. In 2004, Xiao et al. constructed the molecular junctions of three cysteine peptides and measured the conductance upon the environment pH with the STM-BJ technique (Figure 3a). Interestingly, a sigmoid-like titration curve is found for the pH-dependent conductance of the peptides with an amine group when the pH of the solution increases. The maximum change of the single-molecule conductance is at the pH of 7 close to the p*K*_a_ of the amine group measured in the bulk solution, while the conductance of the peptide with an amine and a carboxyl group significantly varies both at the pH close to the p*K*_a_ ≈ 8 and 3. Such pH-dependent conductance is originated from the resultant effect of protonation/deprotonation of the amine and carboxyl groups, which increases the tunneling barrier at high pH and leads to a lower conductance [65]. Similarly, Nichols and collaborators have also proven that the single-molecule junctions of peptide sequence H(EL)_5_C (where H stands for histidine, E for glutamic acid, L for leucine, and C for cysteine) is sensitive to the environment pH with the STM-BJ technique (Figure 3b). The single-molecule conductance decreases from 1.7 nS at pH = 2 to below 0.10 nS at the pH above 6.9. This change is attributed to the H(EL)_5_C bridge which exists in a more compact α-helical state at a low pH, while deprotonation at a high pH leads to the electrostatic repulsion between charged carboxylate groups of glutamate residues, promoting more extended conformation [26]. These discoveries laid a foundation for the application of peptides in pH single-molecule sensors.

Except the biological molecules, supramolecules [27,66,67] have also been proven for the design of single-molecule pH sensors. For example, cucurbit[n]uril (CB[n]) is commonly used due to its unique properties of good biocompatibility. In 2020, Ai et al. used Cucurbit[7]uril (CB[7]) as the host molecule to measure the conductivity properties of CB[7] and its host–guest complex melphalan@CB[7] (Mel@CB[7]) at different pH with STM-BJ technique [27]. They performed conductance measurements in PB solutions at pH = 1, 4, and 7, respectively, and found that the conductance decreased with increasing pH (Figure 3c). At the same pH, the conductance value of CB[7] is larger than that of Mel@CB[7]. This arises from more protons interacting with the carbonyl group of CB[7] in acidic conditions, which can enhance the electron transfer in these molecular junctions, while the addition of Mel decreases the bonding stability of CB[7] to the gold electrode. This provides a new idea for designing pH single-molecule sensing. These results prove a pH-responsive host–guest system for single molecule detection through single-molecule conductance measurements.

Recently, the molecular junctions with nitrogen heterocyclic molecules, such as pyridine [28] and spiropyran derivatives [68], have also been proven to be pH responsive because the nitrogen atom in these molecules can (de)protonate under acidic (basic) conditions. This protonation or deprotonation can significantly change the electronic structure of the molecules and mechanism of charge transport, resulting in conductance switching. For example, in Figure 4a, Tang et al. reported that a molecular junction containing the pyridine nitrogen constructed by STM-BJ could interact with cationic reagents of trifluoro-methanesulfonate (MeOTf) to form protonated pyridinium, displaying a conductance increase of more than one order of magnitude [28]. With a flicker noise analysis, it is interestingly found that the electron tunneling is primarily through space changes to through bond for the protonated pyridinium. Theoretical analysis shows that this protonation leads to the interchange of frontier orbitals and converts destructive quantum interference (QI) into constructive QI for electron transport in the molecular junctions.

Similarly, other nitrogen heterocyclic molecules, imidazole [69,70] and pyrazole [71,72] have been proven as attractive molecules for forming molecular junctions. They can also provide pH-activated connections between Au electrodes and molecules. As shown in Figure 4b, Kamenetska and co-authors investigated the binding mechanism of imidazole in the molecular junctions with an Au electrode by STM-BJ [29]. They measured the conductance of imidazole in solutions with different pH values ranging from 3, 7, 9 to 12. Interestingly, it is found that the conductance peak between 10^−2^ G_0_ and 10^−1^ G_0_ can only appear in basic solutions when the pH is larger than 7. This determines that this molecule bridges the electrodes in its deprotonated form, providing a type of molecular material for single-molecule pH sensing.

In addition to the molecular backbones, pH-sensitive anchoring groups in molecular junctions can also respond to the external stimuli of environmental pH. For instance, pyridine as one of the most used anchoring groups can be protonated to a cation in acidic environments. Brooke et al. used electrochemical STM-BJ techniques to construct the molecular junctions of Ni|4,4′-vinylenedipyridine (4,4′-VDP)|Ni and measured the conductance at different pH and applied potential (Figure 5a) [30]. It was found that the molecular junctions changed from a high conductive state at a high pH of the solution or positive potentials of electrodes, to a low conductive state at a low pH of the solution or negative potentials of electrodes, vice versa. These arise from the pH or potential induced protonation of two pyridyl moieties in the molecules. Furthermore, the relationship between the pH and potential for protonation occurs is obtained by statistically counting the conductance versus distance traces for determining the relative probability of high (*P*_high_) and low (*P*_low_) conductive states. As shown in Figure 5b, the potential E’ versus MSE for *P*_high_/*P*_low_ = 1 against the pH shows a good linear relation. The gradient of the fitted line δ can be used to calculate the charge retained by a protonated 4,4′-VDP molecule adsorbed to Ni electrodes. These dual-response molecular junctions upon the pH and potential can be not only applied in a pH-sensitive switch, but also prove a prototype of a three-terminal sensor with inputting the gate potential to determine the local pH.

The pH-sensitive carboxylic acid group is also one of the most used anchoring groups for forming molecular junctions [73,74,75]. Single-molecule conductance measurement reveals its binding mechanism based on the -COO^−^-Au bond. Thus, the formation probability of molecular junctions strongly relies on the population of deprotonated carboxylic acid molecules, which provides a unique platform for designing a single-molecule pH sensor. As shown in Figure 6a, Zhou and co-authors used the STM-BJ techniques to probe the acid−base chemistry of SAM of 4-(methylthio)benzoic acid (4-MTBA) on the Au(111)/aqueous solution interface [31]. With changing the pH of solution from 0 to 5, the conductance peaks at about 10^−2.90^ G_0_ ascribed to the formations of single-molecule junctions become intense, shown in Figure 6b. The normalized peak intensity versus pH fits well in a sigmoidal curve, due to the increased dissociation of carboxyl groups in less acidic solutions. Furthermore, the quantitative analysis of the conductance peak intensity is used to estimate the interfacial p$K_{a}^{surf}$ value. The fractional surface coverage (θ) ratio of θ_–COO−_ /θ_–COOH_ is proportional to (*I* − *I*_min_)/(*I* − *I*_max_), where *I* is the normalized intensity of conductance peak at a pH, *I*_min_ and *I*_max_ are the minimum and maximum at the current range of pH, respectively. With the Henderson–Hasselbalch type equation, the log[(*I* − *I*_min_)/(*I* − *I*_max_)] versus pH is found to be a good linear relation, shown in Figure 6c. A similar phenomenon can also be observed at the molecular junctions of terephthalic acid (TPA) and 3-methylthiopropionic acid (MPA). In addition, the interfacial p$K_{a}^{surf}$ of MTBA can be also quantitatively evaluated at 6.6, comparable to the reported 7.0 for of 4-mercaptobenzoic acid (4-MBA) immobilized on an Au surface. Therefore, this work advances the application of break junction techniques in interfacial acid−base chemistry at the single-molecule level and provides a feasible way to design the single-molecule pH sensor.

In this section, we can see that pH is initially used as one of the most-used stimuli to tune the fundamental charge transport processes in molecular junctions. Through the protonation or deprotonation of the molecular backbone and anchoring groups, the molecular electronic structures and energy alignment between the molecules and metal electrodes can be significantly altered, as well as the electron transport path. These lead to a variation of single-molecule conductance. Meanwhile, the underlying mechanisms of pH-tuning have also been explored and revealed in these molecular junctions. These enable new understanding of the structure–function relationships of molecular materials for the design of single-molecule devices based on pH modulation, which in turn could enable single-molecule conductance measurement pH sensing.

## 4. Ion Detection

Appropriate amounts of some metal and non-metallic ions dissolved in aqueous media play important roles in the metabolism of plants, animals and humans. However, high concentrations of these ions can lead to many adverse health effects [76]. In addition, some ions, such as Hg^2+^, Cd^2+^, Pb^2+^, and As^3+^, are toxic, which can cause serious debilitating illnesses [77,78,79]. Therefore, it is significantly important to develop highly selective and sensitive methods for detecting ions [80,81,82]. In the following section, we highlight the principles and strategies used in break junction measurements for qualitative and quantitative detection of ions at a single-molecule level [1,2,10,11,15].

An early break junction experiment for the prototype of metal ion detection was carried out by Tao and co-authors in 2004 [83]. They used STM-BJ to construct single-molecule junctions of peptide and measured the conductance and I–V characteristics with Au electrodes. Upon metal ions of Cu^2+^ or Ni^2+^ in the solution, the peptide can be a host for the metal ion guest (Figure 7a). Due to the specific binding of peptides with the metal ions through deprotonated peptide bonds, it significantly changes the configuration of the molecular junctions and thus increases the tunneling current. Therefore, the electrical characteristics difference in the molecular junctions before and after the binding of metal ions paves a way to study the molecular recognition of metal ions at a single-molecule level.

Another host−guest strategy based on the molecular junctions of crown ether is also used for metal ion recognition. In 2020, Yan et al. designed and synthesized a conjugated oligo-(phenyleneethynylene)(OPE) molecule with the a substituted 15-crown-5 ether moiety at the central benzene ring (compound 1) [32]. The rigid and well-defined OPE backbone can suppress the conformational distortion, and the crown ether moiety can coordinate with various alkali metal cations (Figure 7b). With the presence of Li^+^, Na^+^, K^+^, or Rb^+^ in solutions, the single-molecule conductance measurements by using STM-BJ, it is clearly found different step features in the conductance curves in Figure 7c. Furthermore, a good linear relation is found between the conductance values and the ionic effective charge (ze/r) for Li^+^, Na^+^ and Rb^+^ (Figure 7d). While the conductance value of the molecules coordinated to K^+^ does not follow the trend, it is approximately improved 4-fold larger than that in blank. Controlled experiments with OPE-based 18-crown-6 derivative (compound 2) and DFT calculations reveal that a 2:1 sandwich-type supramolecular junction is formed for K^+^, which leads to increase the conductance. Hence, these findings not only advance the understanding of molecule-metal ion interaction and their electron transport in crown ether molecules, but also provide a unique opportunity to develop ion-induced conductance switching and single-molecule sensing device.

Except for metal ions [84], a single-molecule sensor for the non-metal ion of fluoride has also been proposed based on the Lewis acid–base interactions of boron–fluoride coordination in molecular junctions [85]. As shown in Figure 8a,b, two types of organoborane molecular junctions have been successfully constructed by the STM-BJ techniques with Au electrodes. With the present of fluoride ions in solution, a covalent B–F bond can be formed due to the strong Lewis acid–base interactions, which break the original boron-containing-conjugated system. This can change the tunneling mechanism from LUMO to HOMO for electron transport through the 2,5-dimesitylboryl group disubstituted OPE molecules, thus an about four times lower conductance value is observed in the conductance histograms (Figure 8c). Such an organofluoroborate can generate a destructive quantum-interference effect in the dithienoborepin (DTB) molecular junctions, which leads to a conductance switch ratio up to four orders of magnitudes [33]. The significant conductance variation before and after capturing a fluoride ion shows a promising potential application in the design of single-molecule sensors.

Toward a practical single-molecule ion sensor, it is crucial to move from the above-mentioned ion recognition to accurately and quantitatively determine the target analytes. Recently, Hong and co-authors reported a single-molecule conductance ratiometric strategy for quantitatively determining Ag[I] and nicotinamide adenine dinucleotide (NADH) [34]. As illustrated in Figure 9a, the 3,3 ′,5,5 ′-tetramethylbenzidine (TMB) is used as the molecular probe for two reasons: (1) TMB can be oxidized to oxTMB by Ag [I], while oxTMB can be reduced to TMB by NADH; (2) single-molecule break junction experiments clearly show that the conductance of TMB (114.6 nS) is approximately 13 times that of oxTMB (8.7 nS), showing significant conductance peak difference to serve as a ratiometric conductance probe. In the presence of different concentrations of Ag[I] or NADH, the relative proportions of their peaks change accordingly. By counting both the single-molecule conductance traces, the ratio of TMB/oxTMB against logarithmic Ag[I] concentration shows good linearity with a coefficient of 0.99 in Figure 9b. The limit of detection is estimated at an attomolar (aM) level of about ∼34 aM in Figure 9c. Furthermore, the TMB/oxTMB molecular sensing platform is also successfully applied in the quantification of Ag[I] and NADH in the real samples of lake water and cell fragmentation fluid. Therefore, the proposed conductance ratiometric approach based on the break junction approach opens a new way to realize single-molecule sensors with high selectivity and ultra-sensitivity.

In this section, we can see that the qualitative and quantitative detections of metal and non-metal ions can be realized with the single-molecule conductance measurements by the STM-BJ technique. This sensing strongly relies on the interaction between the ions and molecules, such as chelation, Lewis acid–base interactions and redox reaction. In addition, the single-molecule conductance should show a distinct difference before and after adding ions. However, this strategy has good selectivity for detecting specific ions. The unique advantages of high sensitivity and small amounts of sample make the single-molecule conductance measurements an ideal platform for trace analysis.

## 5. Genetic Materials Detection

Nucleotides are important genetic materials in all living organisms. With unique structural and self-assembly properties [86,87,88], they have also been widely studied in biological functions and biotechnological applications. Since the inventions of break junction techniques, the electrical properties of oligonucleotides, including DNA [89,90], RNA [91] and RNA: DNA Hybrids [92], have been explored in the molecular junctions. It was found that the single-molecule conductance of these oligonucleotides is very sensitive to the molecular length [93], the helical conformation [94], the base sequence [95], and the environmental surroundings [96]. For instance, a single base pair mismatch in the short double stranded DNA (dsDNA) can cause the single-molecule conductance changing up by one order of magnitude [97]. The conductance value of guanine-rich RNA:DNA hybrids are is ∼10 times larger than that of dsDNA duplexes with identical sequences [98]. Single-molecule conductance distributions of dsRNA are an order of magnitude wider than DNA:RNA hybrids [99]. These pioneer engineering studies with break junction techniques not only allow people to obtain biological and chemical information of the nucleotide molecules at a single-molecule level, but also have laid a foundation for the development of single-molecule sensors based on genetic materials, as well as for individual biomolecule detection [100].

One proof-of-concept single-molecule sensor based on genetic materials of DNA base pairs for detecting hydrogen-bonding was proposed by Lindsay and co-authors in 2009 [35]. As shown in Figure 10a, a gold STM probe functionalized with a DNA base was brought into contact with a monolayer of nucleosides on a gold surface. When reaching the setpoint of tunneling current, the probe is retracted, and the tunneling current is recorded at the same time. It is interestingly found that the tunnel-decay signals of molecular junctions held together through three hydrogen bonds (guanine–deoxycytidine, G–C, and 2-aminoadenine–thymidine, 2AA–T) in Figure 10b lasts longer than those held together through two (adenine–thymidine, A–T, and 2-aminoadenine–deoxycytidine, 2AA–C) in Figure 10c. The slow decay of the current against the evacuation distance can be attributed to the different role of mechanical interactions through the hydrogen bonds in bases. Furthermore, a mechanical model of the tunnel gap based on conducting atomic force microscopy (CAFM) measurements and theoretical calculations is proposed to estimate the stiffness of hydrogen bonds in the molecular junctions (Figure 10d). Therefore, the sensitivity of the STM approach curve measurements offers a platform for identifying hydrogen bonding with a significant chemical sensitivity.

Similarly, Lindsay and co-authors also used the electron tunneling in molecular junctions to recognize a single base flanked by other bases in short DNA oligomers [36]. As shown in Figure 11a, the STM tip and substrate functionalized with 4-mercaptobenzamide, the hydrogen-bond donor sites of the nitrogen and one hydrogen-bond acceptor site of carbonyl can provide four likely binding modes to the four bases in Figure 11b. During the process of the STM probe randomly drifting over the samples, the DNA nucleotide of d(CCACC) oligomer in the solution can be trapped in the tunneling gap to form molecular junctions through hydrogen bonding, and thus the characteristic bursts of spikes can be observed in the recorded current curves (Figure 11c). Typically, the low-frequency, large-amplitude pulses are assigned to a C base, while the high-frequency, small-amplitude pulses are assigned to a base. Quantitative analysis through counting burst signals of the frequency and amplitude can detect DNA oligomers composed of A, C and ^m^C(5-methyl-deoxycytidine) bases. A furthermore dynamic force spectroscopy by AFM proves that pulling the DNA through the junction with a force of tens of piconewtons would yield reading speeds of tens of bases per second.

Detecting and identifying a genetic material in a real sample with the break junction techniques has also been proven. Hihath et al. used STM-BJ to detect and identify a specific region of messenger RNA (mRNA) from pathogenic bacterial strains of Escherichia coli (*E. coli.*), which partially encodes for the Shiga toxins [37]. As schematically shown in Figure 12a, a 15-base pair (bp) mRNA sequence (*E. coli*: O157:H7 EDL933) bound to a complementary DNA was measured by the STM-BJ. Statistical analysis of conductance histograms with a Gaussian fitting yields the most probable conductance value for this 15 bp hybrid, which is 2.7 × 10^−4^ G_0_. Conversely, there are no detectable peaks in the conductance range when no molecules or only single-stranded DNA probes are present. In addition, several sequences with only one base difference compared to the target mRNA were also measured with the same DNA probe; their corresponding conductance peaks are significantly different, which proves the very good specificity and extremely sensitivity of this method. Furthermore, the limit of detection (LOD) is also explored in a complex environment with the hybridized sequence into an approximately nanomolar concentration solution of synthetic, non-complementary 149 bp RNAs. The conductance histograms in Figure 12b obtained in different concentrations of the O157:H7 sequence still show a clear peak at the aM level. The LOD is estimated at 20 aM with a signal–noise ratio of 3 (Figure 12c). With both *E. coli* O157:H7 and O175:H28 sequences binding to the same DNA probe, two distinguishable peaks for each can also observed in the conductance histogram (Figure 12d). Therefore, this work significantly advances the break junction approach for the development of an electrically based sensor for diagnostic purposes at the single-molecule level.

In this section, we can see that single-molecule conductance measurements can in-situ monitor the biological interaction with high selectivity and sensitivity, which can discover molecular information buried in traditional ensemble experiments, and also offer a pathway to achieve the ultimate goal of realizing single molecule biological detection. In addition, STM-BJ can detect the nucleotide targets in complex biological environments and identify RNA and DNA sequences without a label and cell culture. However, the current reported detections with break junction techniques are limited to the oligonucleotides for which there might be a few reasons: (1) Compared with small organic molecules, nucleotides are large and have complex and changeable structures, which lead to the inability to form distinct conductance peaks. (2) The current amplifiers used in STM-BJ have insufficient sensitivity and are typically limited to 10^−6^ G_0_ in most labs, which cannot detect a very low tunneling current in the biomacromolecules. (3) Constructing a stable molecular junction with the biomacromolecules remains a big challenge. Therefore, there is still a long way to go for employing single molecule conductance measurement to detect biological strains, cancer biomarkers, monitor gene expression and microbial analysis.

## 6. Conclusions and Outlook

In summary, this review focuses on the most recent advancements of single-molecule sensors based on conductance or tunneling current measurements with the STM break junction technique. Based on ion–molecule interactions or reactions, these single-molecule electrical measurements show a great capacity for the qualitative identification and quantitative detection of metal and non-metal ions, and the LOD can reach an aM level; based on the pH-sensitive molecule backbone and anchoring groups, the environmental and interfacial pH can be probed and quantitatively estimated in the molecular junctions; Based on H bonding and base paring in oligonucleotide, the tunneling current or single-molecule conductance measurements can be used to recognize a single base in DNA oligomers and identify the RNA sequence, as well as constructing a single-molecule H-bond sensor based on genetic materials of DNA. Therefore, the single-molecule measurements by break junctions provide a prototype of an electrical sensor with unique advantages, such as simplicity, portability, ultrahigh sensitivity, excellent selectivity, and label-free real-time electrical detection, in a nondestructive manner.

However, the examples detailed in this review are a proof of concept proved in the laboratory research. There is still a long way to go to realize a real single-molecule device for sensing. First, the main problem is that the application for actual single-molecule sensors is inhibited in a break junction setup because the molecular junction in the two-terminal architecture of STM tip and substrate is so sophisticated that it is more commonly used for fundamental science rather than actual applications. Second, constructing robust and reproducible junctions with high yield fabrication and integration for market application remains unachieved. Typically, the formation of molecular junctions requires anchoring groups binding to the two electrodes, thus it is unfavorable for the detection of target molecules without interaction with the electrode or the molecules in the molecular junctions. On the other hands, the real molecular configurations in the molecular junctions can be variable and are hard to identify. Thus, using the tunneling current, single-molecule conductance value and peak as the “fingerprint” for chemical detections should check carefully with controlled experiments, but for real samples, there might be many interferences that will cause signal distortion, as well as decreasing detection efficiency.

To solve the above-mentioned issues, we can use the combination of advanced nanofabrication techniques to efficiently construct a molecular device with high stability, such as developing novel molecular devices using carbon electrodes of SWNTs and graphene [15,101,102]. The combination of various characterization techniques, such as vibrational spectroscopies can help to in situ monitor the molecular structures in the junctions, such as integrating the surface-enhanced Raman spectroscopy in the STM-BJ and MCBJ [19,103,104,105,106]. Therefore, we believe that the break junction measurements toward molecular electronics have a bright future through a good collaboration between chemists, physicists, materials scientists and engineers, and it can greatly complement traditional electronic devices and applications in various fields, including sensing.

## Figures and Tables

**Figure 1 biosensors-12-00565-f001:**
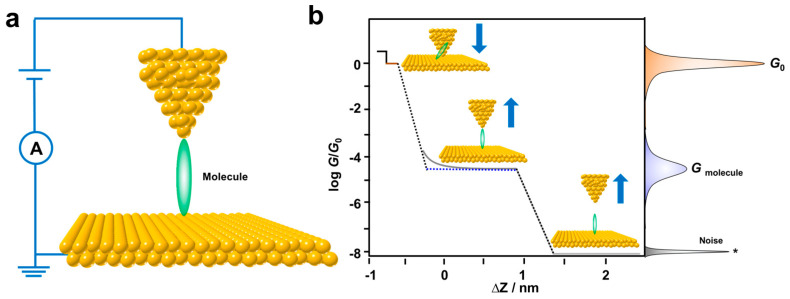
(**a**) Schematic diagram of STM-BJ measurements. (**b**) Simplified schematic representation of a measured conductance trace recorded in the evolution of a molecular junction and 1D conductance histogram on the right side of panel.

**Figure 2 biosensors-12-00565-f002:**
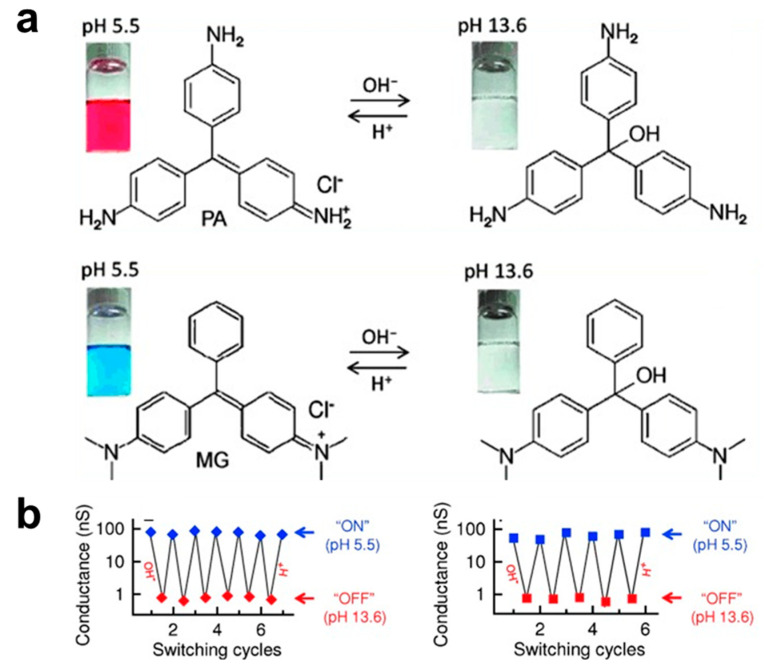
(**a**) Molecular structure and color changes of pararosaniline (PA) and malachite green (MG) at different pH of solutions. (**b**) pH-induced continuous switching of PA (left panel) and MG (right panel) between high conductive (on) and low conductive (off) states. Reprinted with permission from [25]. Copyright 2013 Wiley.

**Figure 3 biosensors-12-00565-f003:**
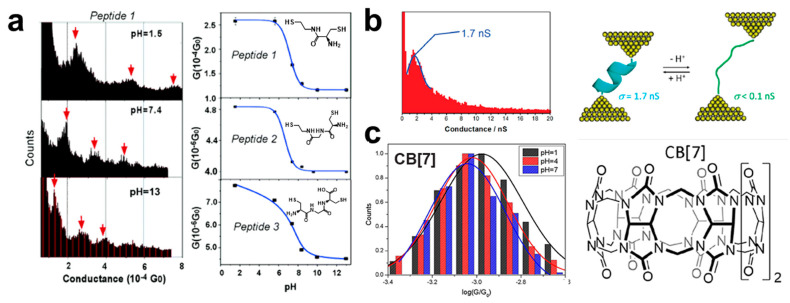
(**a**) The conductance histograms (left panel) and the conductance values against pH for peptides 1, 2, 3 (right panel). The insets are the molecular structures of the three peptides. (**b**) Structural formula and conductance histograms of peptide H(EL)_5_C at pH = 2. (**c**) Single-molecule conductance of CB[7] in PB buffer solution at pH 1, 4 and 7 (left panel), and molecular structure of CB[7] (right panel). Reprinted with permission from [26,27,65]. Copyright 2004 American Chemical Society and 2011 American Chemical Society and 2020 Frontiers.

**Figure 4 biosensors-12-00565-f004:**
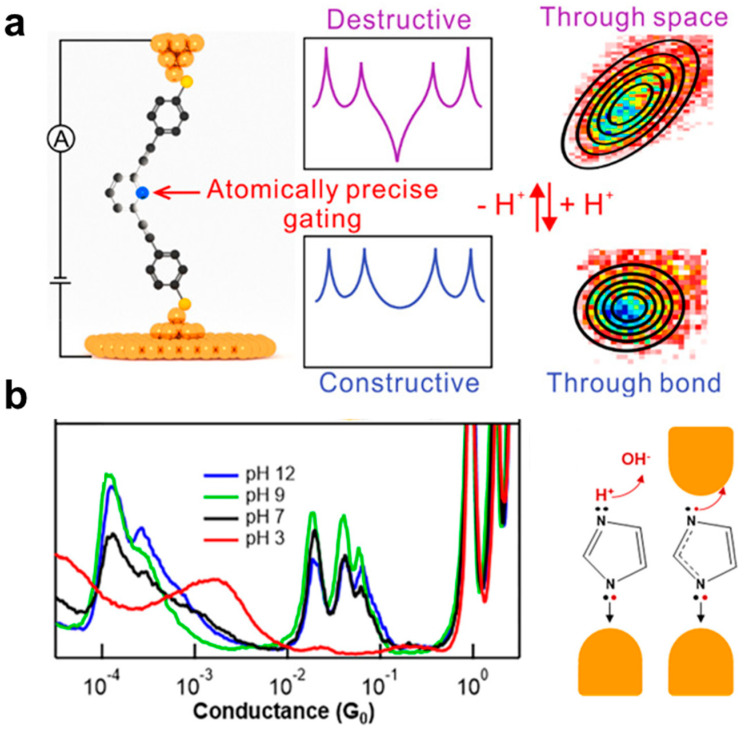
(**a**) Schematic diagram of molecular junction containing the pyridine nitrogen that could interact with cationic reagents. (**b**) The 1D conductance histograms of imidazole measured in aqueous solutions with different pH (left panel), and the mechanism of pH-activated imidazole binding to gold (right panel). Reprinted with permission from [28,29]. Copyright 2021 American Chemical Society and 2020 American Chemical Society.

**Figure 5 biosensors-12-00565-f005:**
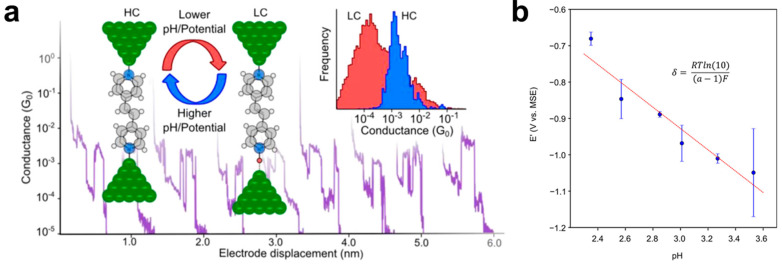
(**a**) Schematic diagram of the STM-BJ technique for probing the conductance of Ni|4,4′-vinylenedipyridine|Ni (4,4′-VDP) at different pH of solution or different potentials of electrodes. The inset represents the two different conductance states of 4,4′-VDP. (**b**) The plot of potential E’ versus MSE for *P*_high_/*P*_low_ = 1 against the pH. Reprinted with permission from [30]. Copyright 2018 American Chemical Society.

**Figure 6 biosensors-12-00565-f006:**
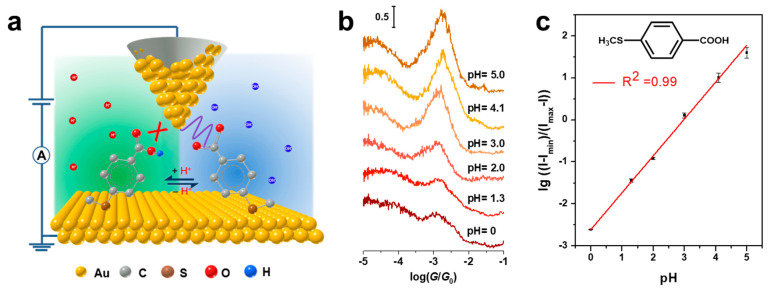
(**a**) Schematic diagram of the STM-BJ technique for probing the acid−base chemistry of 4-MTBA at the liquid−solid interface. (**b**) Normalized intensity of the conductance peak at different the pH of the solution. (**c**) The degrees of dissociation of surface-immobilized -COOH groups derived from the conductance peak intensity vs the pH of the solution. Reprinted with permission from [31]. Copyright 2020 American Chemical Society.

**Figure 7 biosensors-12-00565-f007:**
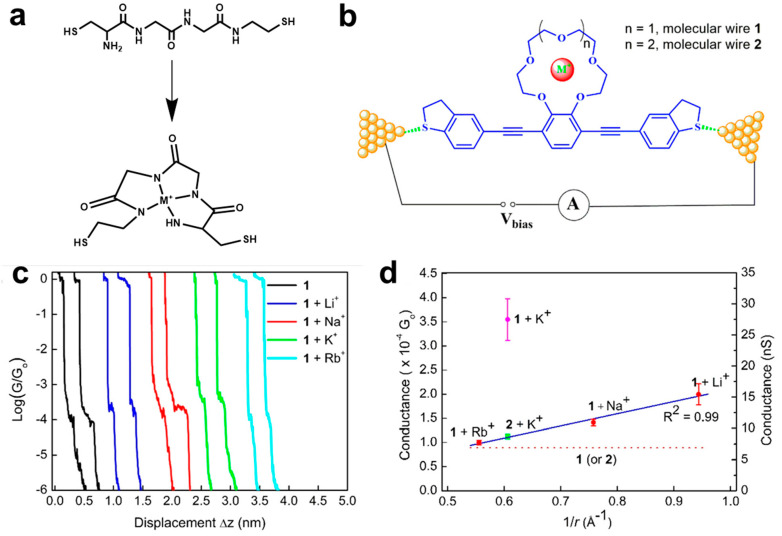
(**a**) The molecular structures of polypeptides before and after coordinating with metal ions. (**b**) Schematic diagram of molecular junctions of OPE molecule with a substituted 15-crown-5 ether (wire 1) or 18-crown-6 (wire 2) moiety for detecting alkali metal ions. (**c**)Typical semilogarithmic conductance traces versus relative electrode displacement (Δz), recorded for the benzo 15-crown-5 based molecular wire 1 in the absence and presence of Li^+^, Na^+^, K^+^, or Rb^+^ ions. (**d**) The plot of single-molecule conductance on effective ionic charge (ze/r) for 1:1 complex of 1 with alkali metal ion (black solid line), and for 2:1 complex of 1 with K^+^ (red dashed line). Reprinted with permission from [32]. Copyright 2020 American Chemical Society.

**Figure 8 biosensors-12-00565-f008:**
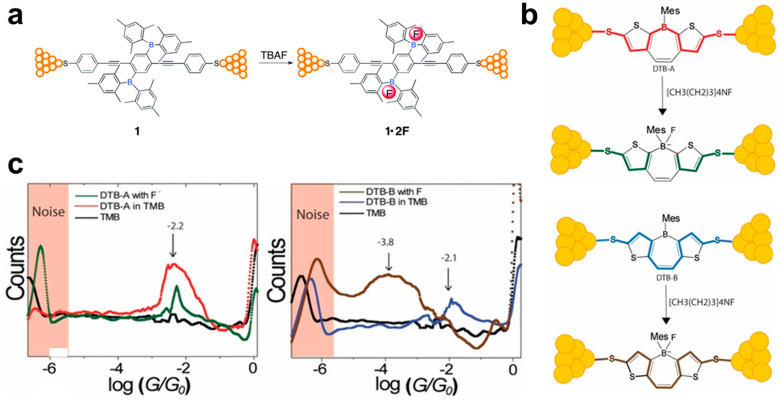
(**a**) Schematic diagram of Lewis acid–base interactions of boron–fluoride coordination in single molecular junctions. (**b**) The conductive pathway in DTB-A (red) and DTB-B (blue) molecular junctions before and after addition tetrabutylammonium fluoride (TBAF). (**c**) The conductance histograms of DTB-A and DTB-B with and without fluoride ions. Reprinted with permission from [33,85]. Copyright 2018 Royal Society of Chemistry and 2019 Wiley.

**Figure 9 biosensors-12-00565-f009:**
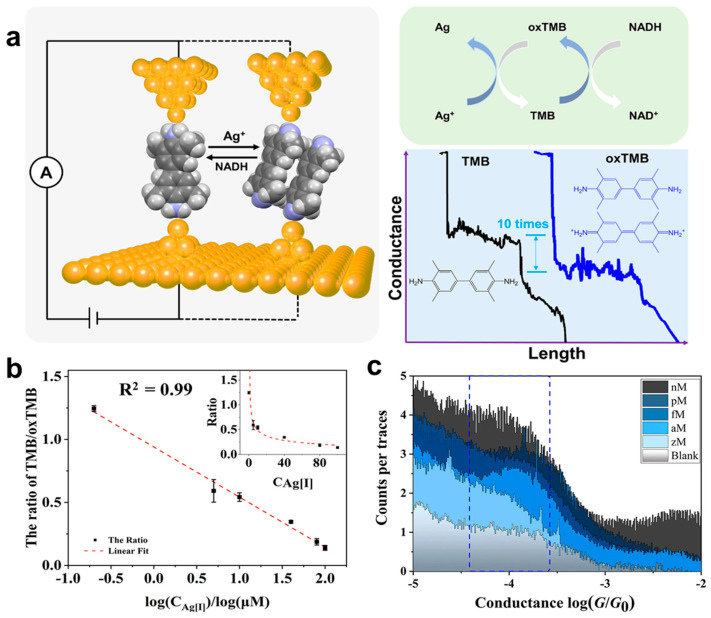
(**a**) Schematic illustration of break junction measurements for Ag[I] and NADH sensing with TMB/oxTMB probe. (**b**) Linear relation between the ratio of TMB/oxTMB and the logarithm of the Ag[I] concentration ranged from 0.2 to 100 μM. The inset shows the ratio of TMB/oxTMB with increasing concentrations of Ag[I]. (**c**) 1D conductance histograms measured at various concentrations (3 zM to 3 nM) of the oxTMB. The blank experiment was measured in the buffer solution. Reprinted with permission from [34]. Copyright 2021 American Chemical Society.

**Figure 10 biosensors-12-00565-f010:**
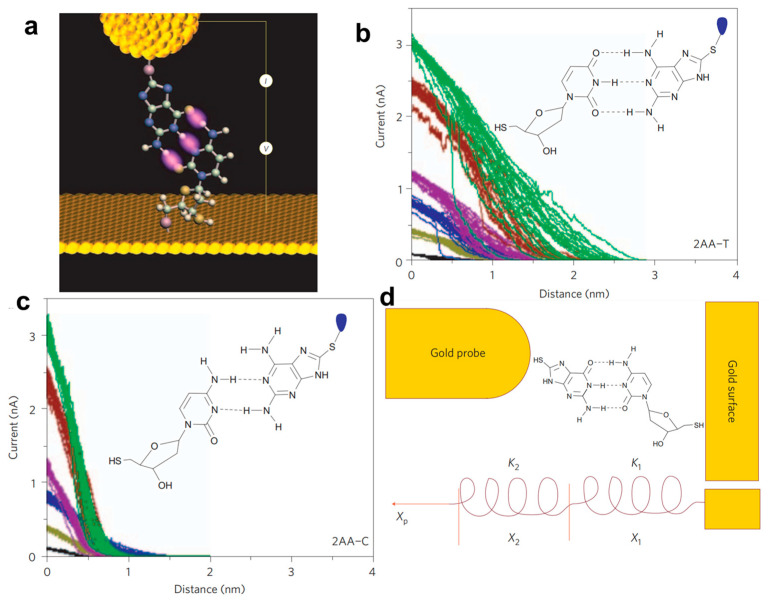
(**a**) Illustration of the STM measurements with a DNA base functionalized Au tip and a monolayer of nucleosides on an Au substrate. The tunneling-current decay curves of (**b**) a 2-amino-8-mercaptoadenine functionalized probe interacting with a thymidine monolayer (2AA–T) and (**c**) a 2-amino-8-mercaptoadenine functionalized probe interacting with a deoxycytidine monolayer (2AA–C). The colors of the curves represent different initial set-point currents from 3 (green), 2.4 (brown), 1.2 (purple), 0.8 (blue), 0.4 (khaki) to 0.1 nA (black). (**d**) The mechanical model illustrates the extension of a molecular junction is accommodated both by the distortion of the molecular system and the junction itself. Reprinted with permission from [35]. Copyright 2009 Nature publishing group.

**Figure 11 biosensors-12-00565-f011:**
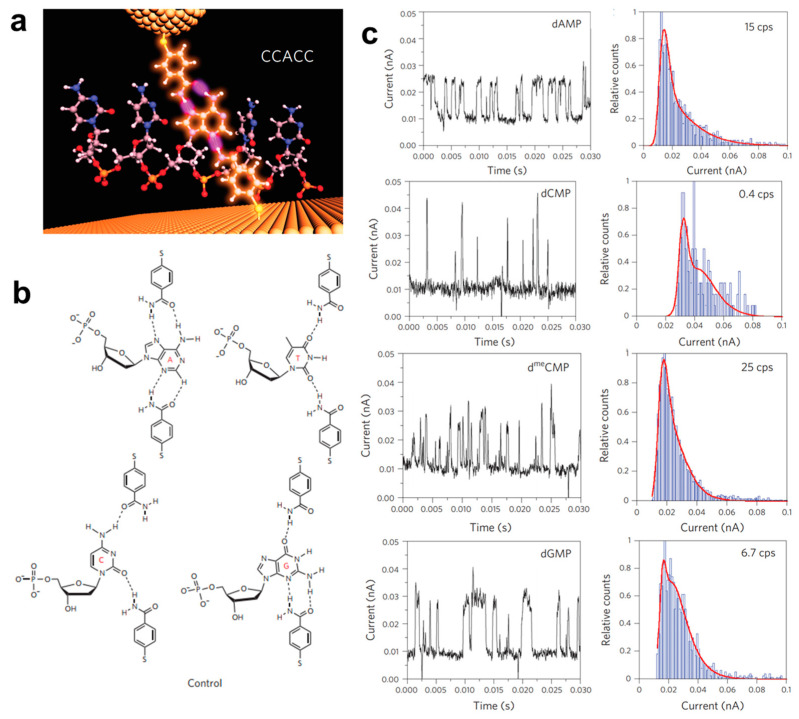
(**a**) Schematic diagram of a molecular junction formed with benzamide groups on the tip and substrate wiring to the single A base in d(CCACC)). (**b**) Proposed hydrogen-bonding modes for the four bases of A, T, C and G. (**c**) Characteristic current spikes produced when nucleotides dAMP, dCMP, d^m^CMP and dGMP were introduced in the molecular junctions, and the corresponding distribution of pulse heights in the right panels. Reprinted with permission from [36]. Copyright 2010 Nature publishing group.

**Figure 12 biosensors-12-00565-f012:**
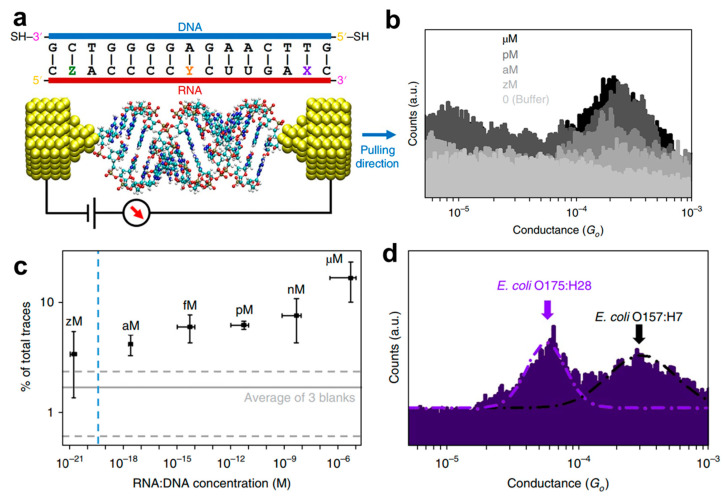
(**a**) Schematic illustration of the break junction experiments on 15 bp RNA: DNA sequences from *E. coli* O157:H7 X  =  A, Y = U and Z = G. (**b**) Representative one-dimensional conductance histograms measured at various concentrations of the O157:H7 sequence. (**c**) Dependence of trace selection on the concentration of target molecules. (**d**) The conductance histograms obtained with both *E. coli* O157:H7 and O175:H28 sequences in the solution. Reprinted with permission from [37]. Copyright 2018 Nature publishing group.

**Table 1 biosensors-12-00565-t001:** The typical examples of molecular junctions constructed with STM-BJ for the three types of sensors.

Sensors	Single-Molecules	Detection Target	Principle	Detection	Ref.
pH detection	Dye molecules	pH = 5.5 or 13.6	Change the hybridization of center C atom	Qualitative	[25]
	Cysteine peptides	pH = 6.9 or 2	Protonation/deprotonation of the amine and carboxyl groups	Qualitative	[26]
	Cucurbit[7]uril	pH = 1, 4, 7, 9	Interaction of proton and carbonyl	Qualitative	[27]
	Pyridine derivatives	No details	Protonation or deprotonation of N atom	Qualitative	[28]
	Imidazole	pH = 3, 7, 9, 12	Protonation or deprotonation of N atoms	Qualitative	[29]
	4,4′-vinylenedipyridine	pH = 2.35,2.57, 2.85, 3.01, 3.26, 3.53	Protonation or deprotonation of N atoms	Qualitative	[30]
	4-(methylthio)benzoic acid	pH = 0~5	Protonation or deprotonation of carboxylic acid	Quantitative	[31]
Ion detection	OPE molecule with 15-crown-5 ether or 18-crown-6	metal ions (Li^+^, Na^+^, K^+^, or Rb^+^)	Host–guest interactions	Qualitative	[32]
	Dithienoborepin	fluoride ion	Lewis acid–base interactions of boron–fluoride	Qualitative	[33]
	3,3′,5,5′-tetramethylbenzidine	Ag[I] (0.2 to 100 μM)	Redox reaction	Quantitative LOD = ∼34 aM	[34]
Genetic materials detection	DNA base pairs	hydrogen-bonding	DNA base pairs for detecting hydrogen-bonding	Qualitative	[35]
	4-mercaptobenzamide	DNA oligomers	Interaction of amino and carbonyl	Qualitative	[36]
	mRNA from Escherichia coli	mRNA (from μM to aM)	Complementary base pairing	Quantitative LOD = ∼20 aM	[37]

## Data Availability

Not applicable.

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
