# Peer review of "Recent Advances in Single-Molecule Sensors Based on STM Break Junction Measurements"

_biosensors, 2022, doi:10.3390/bios12080565_

Round 1

Reviewer 1 Report

In this review, the authors provide an overview of single-molecule sensors with a focus on ion, pH, and nucleotide detections. The authors were able to organize the manuscript very well, highlighting the main aspects of each type of detection, as well as presenting examples and illustrations for each of them. This is an interesting review paper. However, some statements could be clarified for the readers by addressing the following comments.

1. Authors should detail the main advantages and disadvantages related to each detection technique. It is also important to describe the challenges related to the current research. This would improve the manuscript discussion. 

2. More critical analysis and discussions should also be added to the text. 

3. A table could be an interesting tool to summarize all the characteristics and/or the main results/observations obtained with the different types of sensors (i.e., ion, pH, and nucleotide detections). In addition, the analytical performance of the sensors could also be compared and discussed, since the authors commented on the quantitative detection this discussion should be interesting.

4. A section on Future Challenges or Perspectives needs to be added in the review, highlighting the impressions of the authors about the future of the research in this field.

Author Response

Point 1: Authors should detail the main advantages and disadvantages related to each detection technique. It is also important to describe the challenges related to the current research. This would improve the manuscript discussion.

Response 1: According to reviewer’s suggestion, we have supplemented discussions “ Compared with traditional single-molecule detections based on optical, physical and chemical methods, such as fluorescence, surface-enhanced Raman spectroscopy, nanopore, the electrical sensors based on single-molecule conductance measurements can be complementary and have unique advantages with integrating STM technique: (1) It is a label-free real-time electrical detection that requires only a small amount of sample; (2) it is not limited by the working environment. STM can efficiently work in vacuum, atmosphere, solution and other different environment; (3) Dynamic and static molecular information at the interfaces, such as molecular adsorption and surface reaction, can be detected in a real space” and “Compared to ensemble measurements in many traditional analytical methods, single-molecule measure by break junction technique, can not only provide detections with the ultimate resolution, but also are capable to discover the wealth of molecular information that hidden in conventional ensemble experiments. On the other hands, there are still a long way for the applications of the single-molecule conductance measurements in real detecting. The current studies are a proof-of-concept in the laboratory. A brief conclusion and outlook appear in the end.” in the revised Introduction to outline the advantage and challenges of the single-molecule detecting using STM-BJ.

Point 2: More critical analysis and discussions should also be added to the text.

Response 2: We thank the reviewer for the kind suggestion, and have supplemented additional discussions in the three types of sensors as follows:

(1) pH detection: “In this section, we can see that pH is initially used as one of the most used stimuli to tune the fundamental charge transport processes in molecular junctions. Through the protonation or deprotonation of the molecular backbone and anchoring groups, the molecular electronic structures and energy alignment between the molecules and metal electrodes can be significantly altered, as well as changed electron transport path. These lead to a variation of single-molecule conductance. Meanwhile, the underlying mechanisms of pH-tuning have also been explored and revealed in these molecular junctions. These enable new understanding of the structure–function relationships of molecular materials for the design of single-molecule devices based on pH modulation, which in turn could enable single-molecule conductance measurement pH sensing.”

(2) Ion detection: “In this section, we can see that the qualitative and quantitative detections of metal and non-metal ions could be realized with the single-molecule conductance measurements by STM-BJ technique. This sensing strongly relies on the interaction between the ions and molecules, such as chelation, Lewis acid–base interactions and redox reaction. In addition, the single-molecule conductance should show distinct difference before and after adding ions. But this strategy has a good selectivity for detecting specific ions. And the unique advantages of high sensitivity and small amounts of sample make the single-molecule conductance measurements an ideal platform for trace analysis.”

(3) Genetic materials detection: “In this section, we can see that single-molecule conductance measurements can in-situ monitor the biological interaction with high selectivity and sensitivity, which can discover molecular information buried in traditional ensemble experiments, and also offer a pathway to achieve the ultimate goal of realizing single molecule biological detection. In addition, STM-BJ can detect the nucleotide targets in complex biological environments, and identify RNA and DNA sequences without label and cell culture. However, the current reported detections with break junction techniques are limited to the oligonucleotides, there might be few reasons: (1) Compared with small organic molecules, nucleotides are large and have complex and changeable structures, which lead to the inability to form distinct conductance peaks; (2) The current amplifiers used in STM-BJ have insufficient sensitivity and are typically limited to 10-6 G0 in most labs, which cannot detect a very low tunneling current in the biomacromolecules; (3) Constructing a stable molecular junction with the biomacromolecules remains a big challenge. Therefore, there is still a long way to go for employing single molecule conductance measurement to detect biological strains, cancer biomarkers, monitor gene expression and microbial analysis.”

Point 3: A table could be an interesting tool to summarize all the characteristics and/or the main results/observations obtained with the different types of sensors (i.e., ion, pH, and nucleotide detections). In addition, the analytical performance of the sensors could also be compared and discussed, since the authors commented on the quantitative detection this discussion should be interesting.

Response 3: We thank the reviewer for the kind suggestion. We have added a Table 1 to summarize all typical examples of molecular junctions constructed with STM-BJ illustrated in this review for the three types of sensors.

Table 1. The typical examples of molecular junctions constructed with STM-BJ for the three types of sensors.

Sensors

Single-molecules

Detection Target

Principle

Detection

Ref.

pH detection

Dye molecules

pH=5.5 or 13.6

Change the hybridization of center C atom

Qualitative

 [52]

Cysteine peptides

pH=6.9 or 2

Protonation/deprotonation of the amine and carboxyl groups

Qualitative

 [54]

Cucurbit[7]uril

pH=1, 4, 7, 9

Interaction of proton and carbonyl

Qualitative

 [55]

Pyridine derivatives

No details

Protonation or deprotonation of N atom

Qualitative

[58]

Imidazole

pH=3, 7, 9,12

Protonation or deprotonation of N atoms

Qualitative

  [64]

4,4′-vinylenedipyridine

pH=2.35,2.57,2.85,

3.01, 3.26, 3.53

Protonation or deprotonation of N atoms

Qualitative

 [65]

4-(methylthio)benzoic acid

pH=0~5

Protonation or deprotonation of carboxylic acid

Quantitative

  [69]

Ion detection

OPE molecule with 15-crown-5 ether or 18-crown-6

metal ions (Li+, Na+, K+, or Rb+)

Host-guest interactions

Qualitative

 [77]

Dithienoborepin

fluoride ion

Lewis acid–base interactions of boron–fluoride

Qualitative

  [81]

3,3’,5,5’-tetramethylbenzidine

Ag[I] (0.2 to 100 μM)

Redox reaction

Quantitative

LOD=∼34 aM

  [82]

Genetic materials detection

DNA base pairs

hydrogen-bonding

DNA base pairs for detecting hydrogen-bonding

Qualitative

  [98]

4-mercaptobenzamide

DNA oligomers

Interaction of amino and carbonyl

Qualitative

  [99]

mRNA from

Escherichia coli

mRNA (from μM to aM)

Complementary base pairing

Quantitative

LOD=∼20 aM

 [100]

Point 4: A section on Future Challenges or Perspectives needs to be added in the review, highlighting the impressions of the authors about the future of the research in this field.

Response 4: We have outlined the challenges related to the current researches of single-molecule sensing based on single-molecule measurements by break junctions in the Conclusion and Outlook as follows: “The examples detailed in this review are proof-of-concept proved in the laboratory researches. It is still a long way to realize a real single-molecule device for sensing. First, the main problem is that the application for actual single-molecule sensors is inhibited in a break junction setup, because the molecular junction in the two-terminal architecture of STM tip and substrate is so sophisticated that it is more commonly used for fundamental science rather than actual applications. Second, constructing robust and reproducible junctions with high yield fabrication and integration for market application remains unachieved. Typically, the formation of molecular junctions requires anchoring groups binding to the two electrodes, thus it is unfavorable for detection of target molecules without interaction with the electrode or the molecules in the molecular junctions. On the other hands, the real molecular configurations in the molecular junctions can be variable and is hard to identify. Thus, using the tunneling current, single-molecule conductance value and peak as the “fingerprint” for chemical detections should check carefully with controlled experiments, but for real samples, there might many interferences will cause signal distortion, as well as decreasing detection efficiency”.

Reviewer 2 Report

This review focuses on the single-molecule sensors based on STM break junction technique. And their applications in ion detection, pH detection and genetic material detection were reviewed. The advantages, the problems encountered, and possible solutions were all identified. This review is completed in both structure and content. I think it is good for publication in Biosensors. But the following questions should be paid attention before publication.

1.      In the Introduction, it would be better if the authors could highlight the advantages of STM break junction measurement-based single-molecule sensors compared to other types of single-molecule sensors in detail, which would be helpful for understanding the significance of sensors based on STM platform.

2.      New publications should be reviewed and introduced in this paper. Considering this is a review from 2022, some of the research which was detailed introduced was quite old, especially those for pH detection.

Author Response

Point 1: In the Introduction, it would be better if the authors could highlight the advantages of STM break junction measurement-based single-molecule sensors compared to other types of single-molecule sensors in detail, which would be helpful for understanding the significance of sensors based on STM platform.

Response 1: According to reviewer’s suggestion, we have supplemented a discussion “ Compared with traditional single-molecule detections based on optical, physical and chemical methods, such as fluorescence, surface-enhanced Raman spectroscopy, nanopore, the electrical sensors based on single-molecule conductance measurements can be complementary and have unique advantages with integrating STM technique: (1) It is a label-free real-time electrical detection that requires only a small amount of sample; (2) it is not limited by the working environment. STM can efficiently work in vacuum, atmosphere, solution and other different environment; (3) Dynamic and static molecular information at the interfaces, such as molecular adsorption and surface reaction, can be detected in a real space” in the revised Introduction to outline the advantage of the single-molecule detecting using STM-BJ.

Point 2: New publications should be reviewed and introduced in this paper. Considering this is a review from 2022, some of the research which was detailed introduced was quite old, especially those for pH detection.

Response 2: According to the reviewer’s suggestion, we have added citations of 10 papers, especially a few paper (for example, J. Am. Chem. Soc. 2021, 143, 9385-9392; Front. Chem. 2020, 8, 736; Nano Lett. 2020, 20, 4687-4692) about pH detection as well as figures and discussions in the revised manuscript as follows: “Except the biological molecules, supramolecules [55-57] has also been proven for the design of single-molecule pH sensors. For example, cucurbit[n]uril (CB[n]) is commonly used due to its unique properties of good biocompatibility. In 2020, Ai et al. used Cucurbit[7]uril as the host molecule to measure the conductivity properties of CB[7] and its host-guest complex melphalan@CB[7] (Mel@CB[7]) at different pH with STM-BJ technique [55]. They performed conductance measurements in PB solutions at pH=1, 4, and 7, respectively, and found that the conductance decreased with increasing pH. At the same pH, the conductance value of CB[7] is larger than that of Mel@CB[7]. This arise from more protons interact with the carbonyl group of CB[7] in acidic conditions, which can enhance the electron transfer in these molecular junctions; while the addition of Mel decreases the bonding stability of CB[7] to the gold electrode. This provides a new idea for designing pH single-molecule sensing. These results prove a pH-responsive host-guest system for single molecule detection through single-molecule conductance measurements.”

Recently, the molecular junctions with nitrogen heterocyclic molecules, such as pyridine [58] and spiropyran derivatives [59], have also been proven pH-responsive. Because the nitrogen atom in these molecules can (de)protonate under acidic (basic) conditions. This protonation or deprotonation can significantly change the electronic structure of the molecules and mechanism of charge transport result in conductance switching. For example, in Figure 4a, Tang et al. have reported a molecular junction containing the pyridine nitrogen constructed by STM-BJ could interact with cationic reagents of trifluoro-methanesulfonate (MeOTf) to form protonated pyridinium, displaying a conductance increasement more than one order of magnitude [58]. With a flicker noise analysis, it is interesting found that the electron tunneling is primarily through-space changes to through bond for the protonated pyridinium. Theoretical analysis shows that this protonation leads to the interchange of frontier orbitals and converts destructive quantum interference (QI) into constructive QI for electron transport in the molecular junctions.”

“Similarly, other nitrogen heterocyclic molecules, imidazole [60, 61] and pyrazole [62, 63] has been proven as attractive molecules for forming molecular junctions. They can also provide pH-activated connections between Au electrodes and molecules. As shown in Figure 4b, Kamenetska and co-authors have investigated the binding mechanism of imidazole in the molecular junctions with Au electrode by STM-BJ [64]. They measured the conductance of imidazole in solutions with different pH values ranged from 3, 7, 9 to 12. Interestingly, it is found that the conductance peak between 10-2 G0 and 10-1 G0 can only appear in basic solutions when the pH is larger than 7. This determines that this molecule bridges the electrodes in its deprotonated form, providing a type of molecular material for single-molecule pH sensing”.
